# DropMax: Adaptive Stochastic Softmax

## Abstract

We propose DropMax, a stochastic version of softmax classifier which at each iteration drops non-target classes with some probability, for each instance. Specifically, we overlay binary masking variables over class output probabilities, which are learned based on the input via regularized variational inference. This stochastic regularization has an effect of building an ensemble classifier out of exponential number of classifiers with different decision boundaries. Moreover, the learning of dropout probabilities for non-target classes on each instance allows the classifier to focus more on classification against the most confusing classes. We validate our model on multiple public datasets for classification, on which it obtains improved accuracy over regular softmax classifier and other baselines. Further analysis of the learned dropout masks shows that our model indeed selects confusing classes more often when it performs classification.

## 1 Introduction

Deep learning models have shown impressive performances on classification tasks. However, most of the efforts thus far have been made on improving the network architecture, while the predominant choice of the final classification function remained to be the basic softmax regression. Relatively little research has been done here, except for few works that propose variants of softmax function, such as Sampled Softmax (Jean et al., 2014), Spherical Softmax (de Brébisson & Vincent, 2016), and SparseMax (Martins & Fernandez Astudillo, 2016). However, they either do not target accuracy improvement or obtain improved accuracy only on certain limited settings.

In this paper, we propose a novel variant of softmax classifier that achieves improved accuracy over the regular softmax function by leveraging the popular dropout regularization, which we refer to as *DropMax*. At each stochastic gradient descent step in network training, DropMax classifier applies dropout to the exponentiations in the softmax function, such that we consider the true class and a random subset of other classes to learn the classifier. At each training step, this allows the classifier to be learned to solve a distinct subproblem of the given multi-class classification problem, enabling it to focus on discriminative properties of the target class relative to the sampled classes. Finally, when training is over, we can obtain an ensemble of exponentially many [1] classifiers with different decision boundaries.

Moreover, when doing so, we further exploit the intuition that some classes could be more important in reducing the confusion between an instance with all others, as they may be more confused with the target class. For instance in Figure 1(a), an instance of class *cheetah* is likely to be more confused with class *leopard* and *jaguar*, than with less relevant classes such as class *poalr bear* or *humpback whale*. Thus, we extend our classifier to learn the probability of dropping non-target classes, for each instance, such that the stochastic classifier can consider classification between confused classes more often than others, as in Figure 1(b).

The proposed adaptive class dropout can be also viewed as stochastic attention mechanism, that selects a subset of classes each instance should attend to in order for it to be well discriminated from any of the false classes. It also in some sense has similar effect as boosting, since learning a classifier at each iteration with randomly selected non-target classes can be seen as learning a weak classifier, which is combined into a final strong classifier that solves the complete multi-class classification problem with the weights provided by the class retain probabilities learned for each input.

---

[1] to number of classes

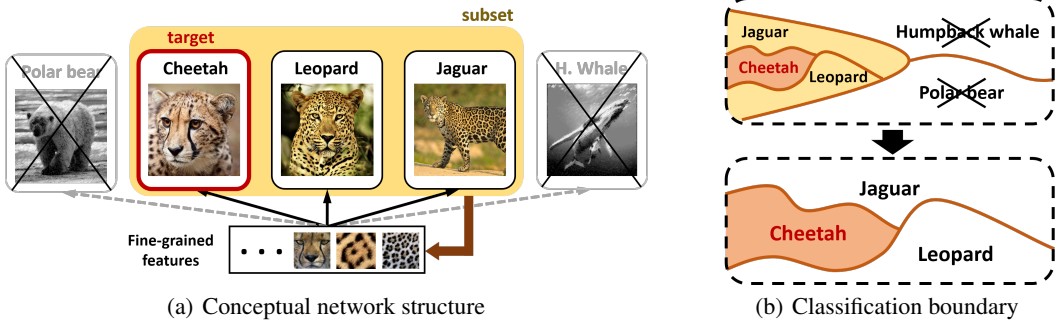

(a) Conceptual network structure      (b) Classification boundary

Figure 1: **concept.** (a) shows conceptual network structure of Dropmax for fine-grained recognition. (b) shows classification boundary of existing models (up) and of Dropmax (bottom).

Our regularization is generic and can be applied even to networks on which the regular dropout is ineffective, such as ResNet, to obtain improved performance. We validate our regularization on multiple public datasets for classification, on which it obtains substantial accuracy improvements.

Our contributions are twofolds:

- We propose a novel stochastic softmax function, DropMax, that randomly drops non-target classes when computing the class probability for each input instance.
- We further propose a variational inference model to learn the dropout probability for each class adaptively for each input, which allows our stochastic classifier to consider highly confused non-target classes more often than others.

## 2 RELATED WORK

**Subset sampling with softmax**    Several existing works propose to consider only a partial susbset of classes to compute the softmax, as done in our work. The main motivation is on improving the efficiency of the computation, as matrix multiplication for computing class logits is expensive when there are too many classes to consider. For example, the number of classes (or words) often exceeds millions in language translation task. Usual practice is to use a shortlist of $30K$ to $80K$ the most frequent target words to reduce the inherent scale of the classification (Bahdanau et al., 2014; Luong et al., 2014). Further, to leverage the full vocabularies, Jean et al. (2014) propose to calculate imporance of each word with a deterministic function and select top-$K$ among them. On the other hand, Martins & Fernandez Astudillo (2016) suggest a new softmax variant that can generate sparse class probabilities, which has a similar effect to aforementioned models. Our model also works with subset of classes, but the main difference is that our model selects non-target classes *randomly* with *learned probabilities* at each iteration with the goal of obtaining an ensemble network in a single training step, for improved classification performance.

**Dropout variational inference**    Dropout (Srivastava et al., 2014) is one of the most popular regularizers for deep neural networks. Dropout randomly drops out each neuron with a predefined probability at each iteration of a stochastic gradient descent, to achieve the effect of ensemble learning by combining exponentially many networks learned during training. Dropout can be also understood as a noise injection process (Bouthillier et al., 2015), which makes the model to be robust to a small perturbation of inputs. Noise injection is also closely related to probabilistic modeling, and Gal & Ghahramani (2015c) has shown that a network trained with dropout can be seen as an approximation to deep Gaussian process. Such Bayesian understanding of dropout allows us to view model training as a posterior process, where predictive distribution is sampled through dropout at test time (Kendall & Gal, 2017). The same process can be applied to convolutional (Gal & Ghahramani, 2015a) and recurrent networks (Gal & Ghahramani, 2015b).

**Learning dropout probability**    In regular dropout regularization, the dropout rate $\rho$ is a tunable parameter via cross-validation. However, some recently proposed models allow to learn the dropout

probability in the training process. Variational dropout (Kingma et al., 2015) assumes that each individual weight has independent Gaussian distribution with mean $\mu$ and variance $\sigma^2$, which are trained with reparameterization trick. Due to the central limit theorem, such Gaussian dropout is identical to the binary dropout, with much faster convergence (Srivastava et al., 2014; Wang & Manning, 2013). Molchanov et al. (2017) show that variational dropout that allows infinite variance results in sparsity, whose effect is similar to automatic relevance determination(ARD). All the aforementioned work deals with the usual posterior distribution not dependent on input at test time. On the other hand, adaptive dropout (Ba & Frey, 2013) learns input dependent posterior at test time by overlaying binary belief network on hidden layers. Whereas approximate posterior is usually assumed to be decomposed into independent components, adaptive dropout allows us to overcome it by learning correlations between network components in the mean of input dependent posterior. Recently, Gal et al. (2017) proposed to train dropout probability $\rho_l$ for accurate estimation of model uncertainty, by reparameterizing Bernoulli distribution with continuous relaxation (Maddison et al., 2016).

## 3 APPROACH

We first introduce the general problem setup. Suppose we have a dataset with $N$ instances,

$$\mathcal{D} = \{(\mathbf{x}_i, y_i) : \mathbf{x}_i \in \mathbb{R}^d, y_i \in \{1, ..., T\}\}_{i=1}^N \tag{1}$$

where $d$ is the data dimension and $T$ is the number of classes to predict. Further suppose a neural network with weight matrices and biases $\boldsymbol{\omega} = \{(\mathbf{W}_1, \mathbf{b}_1), ..., (\mathbf{W}_L, \mathbf{b}_L)\}$ for $L$ layers. The number of neurons for the layer $l = 1, ..., (L-1)$ is $K_1, ..., K_{L-1}$ respectively, and we have $T$ neurons, $\mathbf{o} = (o_1, o_2 ..., o_T)$, at the last layer which are the logits (or scores) for $T$ classes.

As mentioned in the introduction, we propose to randomly drop out exponentiations of output logits, $\exp(o_1), ..., \exp(o_T)$ at the training phase, with the motivation of learning an ensemble of exponentially many classifiers in a single training. Dropout (Srivastava et al., 2014) has been developed with a similar motivation, but our model promotes even stronger diversity among the learned models, by enforcing them to consider different subproblems of the multi-class classification at each stochastic gradient descent step.

To this end, we introduce a dropout binary mask vector $z_t$ with *retain* probability $\rho_t$, which is one minus the dropout probability for each class $t$. $\rho_t$ can be simply set to some predefined probability, such as $0.5$, or can be further learned end-to-end to consider which classes are the most relevant for the correct classification given each instance. In the next subsection, we describe our stochastic version of softmax function, which we refer to as DropMax, along with the method to learn the retain probabilities.

### 3.1 DROPMAX

The original form of the softmax classifier is written as

$$p(y|\mathbf{x}) = \frac{\exp(o_y(\mathbf{x}; \boldsymbol{\omega}))}{\sum_{t=1} \exp(o_t(\mathbf{x}; \boldsymbol{\omega}))}, \tag{2}$$

where $o_t(\mathbf{x}; \boldsymbol{\omega})$ is an output logit for class $t$. One can easily see that if $\exp(o_t(\mathbf{x}; \boldsymbol{\omega})) = 0$, then class $t$ is excluded from the classification and the gradients are not back-propagated from it. From this observation, in *dropmax*, we randomly drop $\{\exp(o_1), ..., \exp(o_T)\}$ based on Bernoulli trials.

$$z_t \sim \text{Bern}(\rho_t), \quad p(y|\mathbf{x}, \boldsymbol{\omega}, \mathbf{z}) = \frac{z_y \exp(o_y(\mathbf{x})) + \varepsilon}{\sum_t z_t \exp(o_t(\mathbf{x})) + T\varepsilon}. \tag{3}$$

where sufficiently small $\varepsilon > 0$ prevents the whole denominator from vanishing. Figure 2 illustrates the contour of this dropmax function with different retain probabilities. However, if we drop the classes based on purely random Bernoulli trials, we may exclude the classes that are important for the classification. Especially, the target class $t^*$ of a given instance should not be dropped, but we cannot manually set retain probabilities $\rho_{t^*} = 1$ since the target classes differ for each instance, and more importantly, we do not know them at test time. We also want the retain probabilities $\rho_1, ..., \rho_T$ to encode meaningful correlations between classes, so that the highly correlated classes may be dropped or retained together.

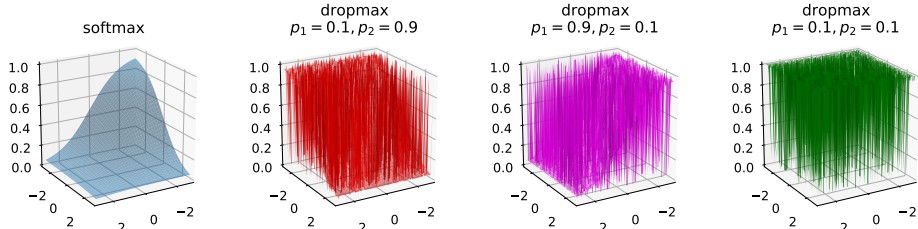

Figure 2: Contour plots of softmax and dropmax with different retain probabilities. For dropmax, we sampled the Bernoulli variables for each data point with fixed probabilities.

To resolve this issue, we adopt the idea of Adaptive Dropout (Ba & Frey, 2013). We let the retain probability to be a function taking the input $\mathbf{x}$. More specifically, we model it as a neural network taking the activations of the last layers of the classification network,

$$\rho_t(\mathbf{x}; \boldsymbol{\nu}) = \text{Sigm}(\mathbf{V}^\top \mathbf{h}_{L-1}(\mathbf{x}) + \mathbf{c}), \tag{4}$$

with additional parameters $\boldsymbol{\nu} = \{(\mathbf{V}, \mathbf{c})\}$. By learning $\boldsymbol{\nu}$, we expect these retain probabilities to be high for the target classes of given inputs, and consider important relations between classes. Based on this retain probability network, the dropmax is defined as follows.

$$z_t|\mathbf{x} \sim \text{Bern}(\rho_t(\mathbf{x}; \boldsymbol{\nu})), \quad p(y|\mathbf{x}, \mathbf{z}; \boldsymbol{\omega}) = \frac{z_y \exp(o_y(\mathbf{x}; \boldsymbol{\omega})) + \varepsilon}{\sum_t z_y \exp(o_t(\mathbf{x}; \boldsymbol{\omega})) + T\varepsilon} \tag{5}$$

The main difference of our model with (Ba & Frey, 2013) is that, unlike in the adaptive dropout where the neurons of intermediate layers are dropped, we drop *classes*. As we stated earlier, this is a critical difference, because by dropping classes we let the model to learn different *(sub)-problems* at each iteration, while in the adaptive dropout we train different *models* at each iteration. Of course, our model can be extended to let it learn the dropout probabilities for the intermediate layers, but it is not our primary concern here. Note that the dropmax can easily be applied to *any* type of neural networks, such as convolutional neural nets or recurrent neural nets, provided that they have the softmax output for the last layer. This generality is another benefit of our approach compared to the (adaptive) dropout that are reported to degrade the performance when used in the intermediate layers of convolutional or recurrent neural networks carelessly.

One limitation of (Ba & Frey, 2013) is the use of heuristics to learn the dropout probabilities that may possibly result in high variance of gradients during training. Instead, we use a continuous relaxation of discrete random variables, which is called concrete distribution (Maddison et al., 2016). It allows us to back-propagate through the (relaxed) bernoulli random variables $z_t$ to compute the gradients of $\boldsymbol{\omega}$ (Gal et al., 2017).

$$z_t = \text{Sigm}\left\{\frac{1}{\tau}\left(\log\frac{\rho_t(\mathbf{x}; \boldsymbol{\nu})}{1 - \rho_t(\mathbf{x}; \boldsymbol{\nu})} + \log\frac{u}{1-u}\right)\right\}, \quad u \sim \text{Unif}(0, 1) \tag{6}$$

The temperature parameter $\tau$ is usually set to $0.1$, which determines the degree of probability mass concentration towards $0$ and $1$. The uniform random variable $u$ is sampled every iteration.

## 4 EFFICIENT APPROXIMATE INFERENCE FOR DROPMAX

In this section, we provide the general learning framework for Dropmax. For notational simplicity, we first define $\mathbf{X}, \mathbf{y}, \mathbf{Z}$ to denote the concatenations of $\mathbf{x}_i, y_i$ and $\mathbf{z}_i$ over all training instances $(i = 1, \dots, N)$. Since our model in (5) involves nonlinear dependencies among variables, the computation of exact posterior distribution $p(\mathbf{Z}|\mathbf{y}, \mathbf{X})$ is intractable, and hence the log-likelihood of our observation $\mathbf{y}|\mathbf{X}$ cannot be directly maximized via exact algorithms such as EM. We instead resort to the standard variational inference where the parameter $\boldsymbol{\theta} = \{\boldsymbol{\omega}, \boldsymbol{\nu}\}$ is optimized with respect to the evidence lower bound (ELBO):

$$\log p(\mathbf{y}|\mathbf{X}; \boldsymbol{\theta}) \geq \sum_{i=1}^{N}\left\{\mathbb{E}_{q(\mathbf{z}_i|\mathbf{x}_i, y_i)}\Big[\log p(y_i|\mathbf{z}_i, \mathbf{x}_i; \boldsymbol{\omega})\Big] - \text{KL}\Big[q(\mathbf{z}_i|\mathbf{x}_i, y_i)\big\|p(\mathbf{z}_i|\mathbf{x}_i; \boldsymbol{\theta})\Big]\right\}. \tag{7}$$

Here, $q(\mathbf{z}|\mathbf{x}, y)$ is a variational distribution to approximate the true posterior $p(\mathbf{z}|\mathbf{x}, y; \boldsymbol{\theta})$. Following the recent advances in the stochastic variational inference, we apply the reparametrization trick in (6) for our model, and the ELBO can be approximated by Monte-Carlo sampling as follows:

$$\frac{1}{S}\sum_{s=1}^{S}\sum_{i=1}^{N}\log p(y_i|\mathbf{z}_i^{(s)}, \mathbf{x}_i; \boldsymbol{\omega}) + \log p(\mathbf{z}_i^{(s)}|\mathbf{x}_i; \boldsymbol{\theta}) - \log q(\mathbf{z}_i^{(s)}|\mathbf{x}_i, y_i), \quad \{\mathbf{z}_i^{(s)}\}_{s=1}^{S} \overset{\text{i.i.d.}}{\sim} q(\mathbf{z}_i|\mathbf{x}_i, y_i).$$
(8)

### 4.1 REGULARIZED VARIATIONAL INFERENCE

Out of several tractable options, we define $q(\mathbf{z}|\mathbf{x}, y)$ in the following simple but practically successful form:

$$q(\mathbf{z}_i|\mathbf{x}_i, y_i) = \prod_{t=1}^{T}\left\{\delta_t(z_{i,t})\mathbb{I}_{\{y_i=t\}} + \mathbb{I}_{\{y_i \neq t\}}p(z_{i,t}|\mathbf{x}_i; \boldsymbol{\theta})\right\}$$
(9)

where $\delta_t(\mathbf{z}_i)$ is the delta function defined as 1 if $t$-th element of $\mathbf{z}_i$ is 1, and 0 otherwise. In other words, we set the variational distribution to have $z_t = 1$ for target class $t$ and to be the same as the prior $p(\mathbf{z}|\mathbf{x}; \boldsymbol{\theta})$ for other classes. In fact, (9) is our deliberate choice in that the corresponding inference coincides with the approximated maximum likelihood estimation, as we will show later. Armed with (9), the KL-divergence term nicely reduces to

$$\text{KL}[q(\mathbf{z}_i|\mathbf{x}_i, y_i)\|p(\mathbf{z}_i|\mathbf{x}_i; \boldsymbol{\theta})] = -\sum_{t=1}^{T}\mathbb{I}_{\{y_i=t\}}\log \rho_t(\mathbf{x}_i; \boldsymbol{\theta}).$$
(10)

so that it encourages the retain probabilities of the target classes to be 1. Now, the ELBO in (8) can be rewritten as

$$\frac{1}{S}\sum_{s=1}^{S}\sum_{i=1}^{N}\log p(y_i|\mathbf{z}_i^{(s)}, \mathbf{x}_i; \boldsymbol{\omega}) + \sum_{i=1}^{N}\sum_{t=1}^{T}\mathbb{I}_{\{y_i=t\}}\log \rho_t(\mathbf{x}_i; \boldsymbol{\theta}), \quad \{\mathbf{z}_i^{(s)}\}_{s=1}^{S} \overset{\text{i.i.d.}}{\sim} q(\mathbf{z}_i|\mathbf{x}_i, y_i). \quad (11)$$

An alternate view to construct (11) is approximating the maximum likelihood estimation. The log-likelihood of observations $\mathbf{y}$ and $\mathbf{z}_t = 1$ (target mask variables) is given by

$$\log p(\mathbf{y}, \mathbf{z}_t|\mathbf{X}; \boldsymbol{\theta}) = \log \sum_{\mathbf{z}_{\backslash t}} p(\mathbf{y}, \mathbf{z}|\mathbf{X}; \boldsymbol{\theta}) = \log \sum_{\mathbf{z}_{\backslash t}} p(\mathbf{y}|\mathbf{z}, \mathbf{X}; \boldsymbol{\omega})p(\mathbf{z}_t|\mathbf{X}; \boldsymbol{\theta})p(\mathbf{z}_{\backslash t}|\mathbf{X}; \boldsymbol{\theta}) \quad (12)$$

where $\mathbf{z}_{\backslash t}$ is the concatenation of mask variables for non-target classes. This quantity can be lower bounded as

$$\log \mathbb{E}_{p(\mathbf{z}_{\backslash t}|\mathbf{X}; \boldsymbol{\theta})}\Big[p(\mathbf{y}|\mathbf{z}, \mathbf{X}; \boldsymbol{\omega})p(\mathbf{z}_t|\mathbf{X}; \boldsymbol{\theta})\Big] \geq \mathbb{E}_{p(\mathbf{z}_{\backslash t}|\mathbf{X}; \boldsymbol{\theta})}\left[\log\Big(p(\mathbf{y}|\mathbf{z}, \mathbf{X}; \boldsymbol{\omega})p(\mathbf{z}_t|\mathbf{X}; \boldsymbol{\theta})\Big)\right] \quad (13)$$

which coincides with (11) if we apply Monte-Carlo approximation for $p(\mathbf{z}_{\backslash t}|\mathbf{X}; \boldsymbol{\theta})$.

Our final ingredient for approximate inference is the regularization to complement crude approximations in (9) especially for non-target outputs (they are simply set as the prior). Following (Gal et al., 2017), we regularize the retain probabilities of non-target classes to have high entropies in order to avoid trivial mapping such as $p(\mathbf{z}|\mathbf{X}; \boldsymbol{\theta}) = \mathbf{1}$ at all times,

$$\Omega(\boldsymbol{\theta}) = -\lambda\sum_{i=1}^{N}\sum_{t=1}^{T}\mathbb{I}_{\{y_i \neq t\}}\Big(\rho_t(\mathbf{x}; \boldsymbol{\theta})\log \rho_t(\mathbf{x}; \boldsymbol{\theta}) + (1 - \rho_t(\mathbf{x}; \boldsymbol{\theta}))\log(1 - \rho_t(\mathbf{x}; \boldsymbol{\theta}))\Big), \quad (14)$$

where $\lambda$ is a hyperparameter to control the amount of regularization. Combining the variational inference objective with the regularization term, our final objective function is

$$\mathcal{L}(\boldsymbol{\theta}) = \sum_{i=1}^{N}\left[\frac{1}{S}\sum_{s=1}^{S}\log p(y_i|\mathbf{z}_i^{(s)}, \mathbf{x}_i; \boldsymbol{\omega}) + \sum_{t=1}^{T}\left\{\mathbb{I}_{\{y_i=t\}}\log \rho_t(\mathbf{x}_i; \boldsymbol{\theta})\right.\right.$$

$$\left.\left.-\lambda\mathbb{I}_{\{y_i \neq t\}}\Big(\rho_t(\mathbf{x}_i; \boldsymbol{\theta})\log \rho_t(\mathbf{x}_i; \boldsymbol{\theta}) + (1 - \rho_t(\mathbf{x}_i; \boldsymbol{\theta}))\log(1 - \rho_t(\mathbf{x}_i; \boldsymbol{\theta}))\Big)\right\}\right], \quad (15)$$

where $\{\mathbf{z}_i^{(s)}\}_{s=1}^{S} \overset{\text{i.i.d.}}{\sim} q(\mathbf{z}_i|\mathbf{x}_i, y_i)$. We optimize this objective function via stochastic gradient descent, where we sample mini-batches to evaluate unbiased estimates of gradients. We found that in practice the single sample ($S = 1$) is enough to get the stable gradients.

### 4.2 PREDICTION

Having trained parameter $\boldsymbol{\theta}$, the prediction for new test input $\mathbf{x}^*$, is given as follows:

$$y^* = \arg\max_y p(y|\mathbf{x}^*; \boldsymbol{\theta}) = \arg\max_y \mathbb{E}_{p(\mathbf{z}|\mathbf{x}^*; \boldsymbol{\theta})}\left[\frac{z_y \exp(o_y(\mathbf{x}^*; \boldsymbol{\omega})) + \varepsilon}{\sum_t z_t \exp(o_t(\mathbf{x}^*; \boldsymbol{\omega})) + T\varepsilon}\right] \quad (16)$$

Evaluating the exact expectation over the dropout masks $\mathbf{z}$ is computationally prohibitive, so we consider the Monte-Carlo approximation as follows:

$$y^* = \arg\max_y \frac{1}{S}\sum_{s=1}^{S}\left[\frac{z_y^{(s)} \exp(o_y(\mathbf{x}^*; \boldsymbol{\omega})) + \varepsilon}{\sum_t z_t^{(s)} \exp(o_t(\mathbf{x}^*; \boldsymbol{\omega})) + T\varepsilon}\right], \quad \{\mathbf{z}^{(s)}\}_{s=1}^{S} \overset{\text{i.i.d.}}{\sim} p(\mathbf{z}|\mathbf{x}^*; \boldsymbol{\theta}). \quad (17)$$

Alternatively, we may approximate the expectation by a simple heuristic where we put the expectation inside the softmax function, as many practitioners do for dropout:

$$y^* = \arg\max_y \frac{\rho_y(\mathbf{x}^*; \boldsymbol{\theta}) \exp(o_y(\mathbf{x}^*; \boldsymbol{\omega})) + \varepsilon}{\sum_t \rho_t(\mathbf{x}^*; \boldsymbol{\theta}) \exp(o_t(\mathbf{x}^*; \boldsymbol{\omega})) + T\varepsilon}. \quad (18)$$

## 5 EXPERIMENTS

**Baselines and our models**   We first introduce relevant baselines and our models.

**1) Base Network.**  The baseline CNN network, that only uses the hidden unit dropout at fully connected layers, or no dropout regularization at all if the network does not have fully connected layers other than the last layers, as in the case of ResNet (He et al., 2016).

**2) Sampled Softmax.** Base network with sampled softmax (Jean et al., 2014). The sampling function $Q(y|\mathbf{x})$ is either uniformly distributed or learned during training. At training time, the number of non-target classes to be randomly selected is set to one of $\{20\%, 40\%, 60\%\}$ of total classes, while target classe is always selected. At testing time, class probabilities are obtained from (2).

**3) Sparsemax.**  Base network with Sparsemax loss proposed by Martins & Fernandez Astudillo (2016), which produces sparse class probabilities.

**4) Deterministic Attention.** The deterministic version of Adaptive Dropmax, where the stochastic $\mathbf{z}$ is replaced with deterministic $\boldsymbol{\rho}$ everywhere. $\lambda$ is found in the same range in Adaptive Dropmax.

**5) Random Dropmax.** Our all-random DropMax, where each non-target class is randomly dropped out with a predefined retain probability $p \in \{0.2, 0.4, 0.6\}$ at training time. At test time, we perform prediction with 30 or 100 random dropouts, and then average out class probabilities to select the class with maximum probability. We also report the performance based on (2), without sampling.

**6) Adaptive Dropmax.** Our adaptive stochastic softmax, where each class is dropped out with input dependent probabilities trained from the data. The entropy scaling parameter $\lambda$ is found among 3 to 5 values such as $\{10^0, 10^{-1}, 10^{-2}, 10^{-3}, 10^{-4}\}$.

For random and adaptive dropmax, we considered two different approaches mentioned in Section 4.2 to compute the dropout masks: 1) using test-time sampling in (17), and 2) using an approximate expectation in (18).

**Datasets and base networks**   We validate our method on multiple public datasets for classification, with different network architecture for each dataset.

**1) MNIST.** This dataset consists of $60,000$ images that describe hand-written digits from 0 to 9. We experiment with varying number of training instances: $1K$, $5K$, and $55K$. The validation and test set has $5K$ and $10K$ instances, respectively. As for the base network, we use the CNN provided in the Tensorflow Tutorial, which has a similar structure to LeNet.

**2) CIFAR-10.** This dataset consists of 10 generic object classes, which for each class has 5000 images for training and 1000 images for test. We use ResNet-34 (He et al., 2016) as the base network, which has 32 Conv layers.

**3) CIFAR-100.** This dataset consists of 100 generic object classes. It has 500 images for training and 100 images are for test for each class. We use the same base network as CIFAR-10.

Table 1: Test classification error (%). The first three columns are results on MNIST dataset. For MNIST, the reported number is median of 3 runs. For other dataset, the reported numbers are one-time runs.

| Models | $1K$ | $5K$ | $55K$ | C10 | C100 | AwA |
|---|---|---|---|---|---|---|
| Base Network | 7.17 | 2.19 | 0.84 | 7.97 | 30.81 | 26.77 |
| Sampled Softmax (uniform $Q$) | 7.48 | 2.17 | 0.91 | 8.29 | 30.52 | 28.04 |
| Sampled Softmax (learned $Q$) | 7.62 | 2.45 | 0.85 | 8.37 | 30.58 | 26.60 |
| Sparsemax | 6.84 | 2.28 | 0.82 | 7.73 | 31.54 | 28.41 |
| Deterministic Attention | 7.15 | 2.17 | **0.70** | 7.64 | 29.60 | 26.85 |
| Random Dropmax (17) | 7.11 | 2.23 | 0.89 | 8.88 | 30.84 | 27.55 |
| Random Dropmax (18) | 7.43 | 2.33 | 0.89 | 8.09 | 30.35 | 26.81 |
| Adaptive Dropmax (17) | **6.56** | 1.96 | 0.77 | 7.49 | **29.56** | **26.40** |
| Adaptive Dropmax (18) | 6.59 | **1.95** | 0.75 | **7.47** | 29.58 | 26.44 |

**4) AWA.** This is a dataset for classifying different animal species (Lampert et al., 2009), that contains $30,475$ images from $50$ animal classes such as *cow*, *fox*, and *humpback whale*. For each class, we used $50$ images for test, while rest of the images are used as training set. The resultant training set is quite imbalanced between classes. We used AlexNet (Krizhevsky et al., 2012) pretrained on ImageNet as the base network.

**Experimental Setup**    Here we briefly mention about the experimental setup for MNIST dataset. The number of iteration is $20k$ with batch size 50. We use Adam optimizer (Kingma & Ba, 2014), with learning rate starting from $10^{-4}$. The $\ell_2$ weight decay parameter is searched in the range of $\{0, 10^{-4}, 10^{-3}, 10^{-2}\}$. All the hyper-parameters are found with a hold-out set. Our model is implemented using Tensorflow Abadi et al. (2016) library, and we will release our codes upon acceptance of our paper, for reproduction.

## 5.1    QUANTITATIVE EVALUATION

**Multi-class classification.**    We report the classification performances of our models and the baselines in Table 1. The results show that the variants of softmax function such as SparseMax and Subset Sampling perform similarly to the original softmax function (or worse). Random dromax also performs similarly. On the other hand, Deterministic-Attention outperforms the baselines on several datasets, and our adaptive DropMax further improves on it with added stochasticity. Such tendency is consistent across all datasets, either when using test-time sampling or the approximation.

Adaptive dropmax obtains higher accuracy gains on CIFAR-100 that has larger number of classes, which is reasonable since with larger number of classes, they are more likely to get confused on which case our stochastic attention-like mechanism becomes helpful. Further, having larger number of classes allows the model to explore larger number of ensembles. On MNIST dataset, we also observe that the adaptive dropmax is more effective when the number of training instances is small, which we attribute to the advantage of variational inference.

**Convergence rate.**    We examine the convergence rate of our model against the base network with regular softmax function. Figure 3 shows the plot of cross entropy loss computed at each training step, on CIFAR-100 dataset. To reduce the variance of $\mathbf{z}$, we plot with $\rho$ instead (training is done with $\mathbf{z}$). Note that despite of sampling process of $\mathbf{z}$, dropmax shows almost the same convergence rate to the base model, and also shows slightly lower test loss, effectively preventing overfitting.

Figure 3: Convergence plot

## 5.2    QUALITATIVE ANALYSIS

We further perform qualitative analysis of our model to see how exactly it works and where the accuracy improvements come from.

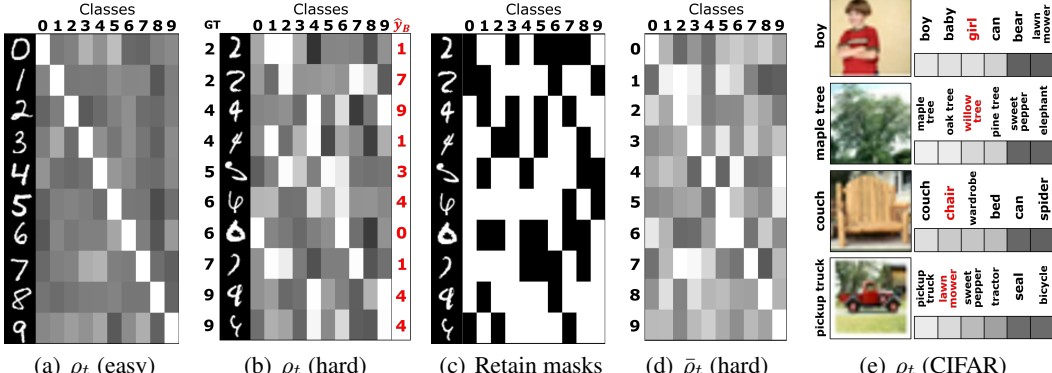

| (a) $\rho_t$ (easy) | (b) $\rho_t$ (hard) | (c) Retain masks | (d) $\bar{\rho}_t$ (hard) | (e) $\rho_t$ (CIFAR) |

Figure 4: Visualization of class dropout probabilities for example test instances from MNIST dataset. (a) and (b) shows estimated class retain probability for easy and difficult test instances respectively, where the last column of (b), $\hat{y}_B$ are the predictions from the baseline model. (c) shows generated retain masks from (b). (d) shows the average retain probability per class for hard instances. (e) shows examples from CIFAR-100 dataset with top-4 and bottom-2 retain probabilities. Red color denotes base model predictions.

Figure 4(a) shows the retain probabilities estimated for easy examples, in which case the model set the retain probability to be high for the true class, and evenly low for non-target classes. Thus, when the examples are easy, the dropout probability estimator works like a second classifier. However, for difficult examples in Figure 4(b) that is missclassified by the base softmax function, we observe that the retain probability is set high for the target class and few other candidates, as this helps the model focus on the classification between them. For example, instances from class 9 set high retain probability for class 4, since their handwritten characters looks somewhat similar to number 9. In general, instances in the same class tend to have similar class dropout pattern (e.g. class 9). However, the retain probability could be set differently even across the instances from the same class, which makes sense since even within the same class, different instances may get confused with different classes. For example, for the first instance of 2, the class with high retain probability is 1, which is somewhat confused with the instance. However, for the second instance of 2, the network set class 7 and 8 with high retain probability as this particular instance looks like 7 or 8.

Similar behaviors can be observed on CIFAR-100 dataset (Figure 4(e)) as well. As an example, for class *boy*, dropmax set the retain probability high on class *boy*, *baby*, and *girl*, which shows that it attends to most confusing classes and enables the model to focus on fine-grained classification.

We further examine the class-average dropout probabilities for each class in MNIST dataset in Figure 4(d). We observe that some classes (3, 5, 9) are more often retained than others, as they often get confused with the other classes, while classes that are easily distinguishable from others, such as classes (0, 1), are retained with very low probability. This asymmetry allows the network to focus more on those most often confused classes, which in turn will enable the network to learn a more accurate decision boundary.

# 6 CONCLUSION

We proposed a stochastic version of a softmax function, DropMax, that randomly drops some non-target classes at each iteration of the training step. DropMax enables to build an ensemble over exponentially many classifiers with high diversity, that provide different decision boundaries. We further proposed to learn the class dropout probabilities based on the input, such that it can consider the discrimination of each instance against more confused classes. We cast this as a Bayesian learning problem and present how to optimize the parameters through variational inference. We validate our model on multiple public datasets for classification, on which our method obtains consistent performance improvements over the base models. Our method also converges fast and incurs marginal extra training cost. Further qualitative analysis shows that it is able to learn dropout mask differently for each class, and for each given instance. Potential future work includes a more accurate approximation of the $q$ function, and extension of the method to multi-task learning.

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
