# OpenReview forum: "DropMax: Adaptive Stochastic Softmax"
_ICLR.cc/2018/Conference — Invite to Workshop Track_

### Official Review · AnonReviewer2 · 2017-11-24
**Adaptively zero out class logits based on the input**

**Rating:** 6
**Confidence:** 3

**Review:**

This paper propose an adaptive dropout strategy for class logits. They learn a distribution q(z | x, y) that randomly throw class logits. By doing so they ensemble predictions of the models between different set of classes, and focuses on more difficult discrimination tasks. They learn the dropout distribution by variational inference with concrete relaxation.

Overall I think this is a good paper. The technique sounds, the presentation is clear and I have not seen similar paper elsewhere (not 100% sure about the originality of the work though).

Pro:
* General algorithm

Con:
* The experiment is a little weak. Only on CIFAR100 the proposed approach is much better than other approaches. I would like to see the results on more datasets. Maybe should also compare with more dropout algorithms, such as DropConnect and MaxOut.

---

> ### Author Response · Authors · 2017-12-23
> **Response to AnonReviewer2**
>
> We really appreciate your comments
>
> - The experiment is a little weak. Only on CIFAR100 the proposed approach is much better than other approaches. I would like to see the results on more datasets. Maybe should also compare with more dropout algorithms, such as DropConnect and MaxOut.
>
> : We are experimenting on ImageNet 1K dataset, and will include the results if we obtain the results by the rebuttal deadline.
>
> DropConnect and MaxOut are not much relevant to our motivation of learning an  ensemble of multiple classifiers in a single training stage, as they do not drop out classes.

---

### Official Review · AnonReviewer1 · 2017-11-26
**A relevant idea, but not especially innovative and not brilliantly carried out.**

**Rating:** 4
**Confidence:** 3

**Review:**

The paper discusses dropping out the pre-softmax logits in an adaptive manner. This isn't a huge conceptual leap given previous work, for instance that of Ba and Frey 2013 or the sequence of papers by Gal and his coauthors on variational interprations of dropout. In the spirit of the latter series of papers on variational dropout there is a derivation of this algorithm using ideas from variational inference. The variational approximation is a bit odd in that it doesn't have any variational parameters, and indeed a further regulariser in equation (14) is needed to give the desired behaviour. A fairly small, but consistent improvement on the base model and other similar ideas is reported in Table 1. I would have liked to have seen results on ImageNet. I don't find (the too small) Figure 2 to be compelling evidence that "our dropmax effectively prevents
overfiting by converging to much lower test loss". The test loss in question looks like a noisy version of the base test loss with a slightly lower mean. There are grammatical errors throughout the paper at a higher rate than would normally be found in a successful submission at this stage. Figure 3 illustrates the idea nicely. Which of the MNIST models from Table 1 was used?

---

> ### Author Response · Authors · 2017-12-23
> **Response to AnonReviewer1**
>
> We really appreciate your comments.
>
> - The paper discusses dropping out the pre-softmax logits in an adaptive manner. This isn't a huge conceptual leap given previous work, for instance that of Ba and Frey 2013 or the sequence of papers by Gal and his coauthors on variational interpretations of dropout.
>
> : The main focus of this paper is not interpreting dropout (or adaptive dropout) wrt variational inference. Those are simply our choice of tools for solving the proposed problem, and the main novelty comes from stochastically ruling out classes from consideration at each iteration. None of the previous work exploits such idea.
>
>
> - The variational approximation is a bit odd in that it doesn't have any variational parameters.
>
> : Our decision of setting the q (or recognition) network the same as the p (or prior) network is motivated from (Sohn et al., 2015) (Section 4.2).  Since we are training with q network while predicting with p network, the consistency between the two network is crucial in obtaining the desired performance. It is indicated by KL[q||p] term in the Eq. (7).
>
> Suppose use a different set of variational parameters \phi for q(z|x,y). The problem in this case is that reconstructing y with q(z|x,y;\phi) and reconstructing y with p(z|x;\theta) are significantly different in their difficulties. The former is much easier because it learns trivial mapping y -> z -> y, where the dimension of z is the same as that of y. Thus, we decided to replace q(z|x,y;\phi) with q(z|x,y;\theta) that shares the same structure and the set of parameters with p(z|x;\theta). In our preliminary experiment, we also experimented with the model that uses a separate parameter for q, but it did not work well.
>
>
> - a further regulariser in equation (14) is needed to give the desired behaviour.
> : Since regularized variational inference is a general framework and allows us to avoid the weird solution all z=0 or z=1, we argue that (14) is reasonable.
>
>
> - I would have liked to have seen results on ImageNet.
> : We will run the experiments on the ImageNet dataset and will include the results in the revision if we obtain the results by the rebuttal deadline.
>
>
> - I don't find (the too small) Figure 2 to be compelling evidence that "our dropmax effectively prevents overfitting by converging to much lower test loss". The test loss in question looks like a noisy version of the base test loss with a slightly lower mean.
>
> : The plot was not the most representative and we included in a more stable version in the revision. Also the main point we want to make with Figure 2 is that our model is still able to achieve lower test loss, while retaining the same convergence speed as the baseline.
>
>
> - There are grammatical errors throughout the paper at a higher rate than would normally be found in a successful submission at this stage.
> : We have corrected the grammatical errors in the revision.
>
>
> - Which of the MNIST models from Table 1 was used?
> : We used the MNIST-1K model.

---

> > ### Comment · AnonReviewer1 · 2018-01-12
> > **Response to rebuttal**
> >
> > Thank you for your response which I have read.

---

### Official Review · AnonReviewer3 · 2017-11-27
**Needs more clarity and a deterministic baseline**

**Rating:** 6
**Confidence:** 4

**Review:**

Pros
- The proposed model is a nice way of multiplicatively combining two features :
  one which determines which classes to pay attention to, and other that
provides useful features for discrimination.

- The adaptive component seems to provide improvements for small dataset sizes
  and large number of classes.

Cons
- "One can easily see that if o_t(x; w) = 0, then class t becomes neutral in the
  classification and the gradients are not back-propagated from it." : This does
not seem to be true. Even if the logits are zero, the class would have a
non-zero probability and would receive gradients. Do the authors mean
exp(o_t(x;w)) = 0 ?

- Related to the above, it should be clarified what is meant by dropping a
  class. Is its logit set to zero or -\infty ? Excluding a class from the
softmax is equivalent to having a logit of -\infty, not zero. However, from the
equations in the paper it seems that the logit is set to zero. This would not
result in excluding the unit. The overall effect would just be to raise the
magnitude of logits across the entire softmax.

- It seems that the model benefits from at least two separate effects - one is
  the attention mechanism provided by the sigmoids, and the other is the
stochasticity during training. Presently, it is not clear if only one of the
components is providing most of the benefits, or if both things are useful. It
would be great to compare this model to a non-stochastic one which just has the
multiplicative effects applied in a deterministic way (during both training and
testing).

- The objective of the attention mechanism that sets the dropout mask seems to
  be the same as the primary objective of classifying the input, and the
attention mechanism is prevented from solving the task by adding an extra
entropy regularization. It would be useful to explain more why this is needed.
Would it not be fine if the attention mechanism did a perfect job of selecting
the class ?

Quality
The paper makes relevant comparisons and is overall well-motivated. However,
some aspects of the paper can be improved by adding more explanations.

Clarity
Some crucial aspects of the paper are unclear as mentioned above.

Originality
The main contribution of the paper is similar to multiplicative gating. The
added stochasticity and the model ensembling interpretation is probably novel.
However, experiments are insufficient to determine whether it is this novelty
that contributes to improved performance or just the gating.

Significance
This paper makes incremental improvements and would be of moderate interest to
the machine learning community.

Typos :
- In Eq 3, the numerator has z_t. Should that be z_y ?
- In Eq 5, the denominator has z_y. Should that be z_t ?

---

> ### Author Response · Authors · 2017-12-23
> **Experimental comparison against a deterministic attention model.**
>
> We really appreciate your comments.
>
> - It seems that the model benefits from at least two separate effects - one is the attention mechanism provided by the sigmoids, and the other is the stochasticity during training. Presently, it is not clear if only one of the components is providing most of the benefits, or if both things are useful. It would be great to compare this model to a non-stochastic one which just has the multiplicative effects applied in a deterministic way (during both training and testing).
>
> : As said, our model benefits from two separate effects - 1) adaptive input-dependant attention generation and 2) stochasticity during training.
>
> The effect of 1) is clear since our adaptive dropmax significantly outperforms random dropmax. To show the effect of 2) we added in the results from the deterministic model in the revision, which we name as Deterministic-Attention, in Table 1. This model is almost identical to “Adaptive-Dropout”, except that the stochastic ‘z_t’ is replaced with deterministic ‘\rho_t’.
>
> We observe that stochasticity does indeed help improve the model performance, as Adaptive-Dropmax outperformed Deterministic-Attention by 0.59% in MNIST-1K, 0.22% in MNIST-5K and similarly on the other datasets (except on MNIST-55K).
> Further, our deterministic attention model has both KL term and entropy regularizer as in “Adaptive-Dropout”, with \lambda found via separate holdout set, such that the target class is strongly attended for each input while non-target classes are not. This design, which is also used in our Adaptive-Dropmax, is also a novelty of our model since a naive implementation of deterministic attention produces much worse results than the base model,
>
>
> - The objective of the attention mechanism that sets the dropout mask seems to be the same as the primary objective of classifying the input, and the attention mechanism is prevented from solving the task by adding an extra entropy regularization. It would be useful to explain more why this is needed. Would it not be fine if the attention mechanism did a perfect job of selecting the class?
>
> : The objective of the dropout mask generator is to stochastically rule out non-target classes such that the model can learn features for both coarse-grained and fine-grained classification. If we allow the dropout mask generator to become another classifier, then the original classifier has no problem to solve and will not learn anything useful, and thus we should differentiate the role of the classifier and the dropout mask generator.
>
> We found that even in the case where it is easy enough for the mask generator to do a perfect job of selecting the target (See Figure 4(a) in the revision - Figure 3(a) in the original paper) the performance was the best when the non-target classes are not completely ruled out as \lambda was found to be nonzero (0.1 ~ 0.0001).
>
> To verify it, we experimented with Deterministic-Attention model, with the Sigm() in Eq. (4) replaced with Softmax(). It makes the mask generator to be another classifier, because generated masks become mutually exclusive, with only one of them close to 1 per each instance. The entropy regularizer (14) is removed for our purpose. We tested it on MNIST and Cifar-100, and the results are as follows:
> MNIST-1K: 7.13
> MNIST-5K: 2.57
> MNIST-55K: 1.09
> Cifar-100: 30.38
> The results are similar to or worse than the baseline, meaning that the role of the mask generator should be controlled in a principled way.
>
>
> - The main contribution of the paper is similar to multiplicative gating. Experiments are insufficient to determine whether it is just the gating that contributes to improved performance.
>
> : As mentioned above, the newly added in experimental results for the deterministic attention model shows that the stochasticity is still important for obtaining meaningful performance improvement, as it enables to obtain an ensemble of exponentially many classifiers in a single model training.

---

> ### Author Response · Authors · 2017-12-23
> **Clarification**
>
> We really appreciate your comments.
>
> - "One can easily see that if o_t(x; w) = 0, then class t becomes neutral in the classification and the gradients are not back-propagated from it." : This does not seem to be true. Even if the logits are zero, the class would have a non-zero probability and would receive gradients. Do the authors mean exp(o_t(x;w)) = 0 ?
>
> : This is indeed correct and is a mistake caused by the explanation of a legacy model. We have experimented with two different versions of Dropmax (one that drops out the o_t and the other that drops out the exp(o_t) and opted to go with the former.
>
> In the revision, we have corrected the inaccurate description of the model and added in new experimental results based on the dropout of the exponential term (including Figure 4). The results show that dropping exp(o_t) =0 yields similar classification errors to dropping o_t=0, except on Cifar-10, on which the former significantly outperforms the latter.
>
>
> - Related to the above, it should be clarified what is meant by dropping a class. Is its logit set to zero or -\infty ? Excluding a class from the softmax is equivalent to having a logit of -\infty, not zero. However, from the equations in the paper it seems that the logit is set to zero. This would not result in excluding the unit. The overall effect would just be to raise the magnitude of logits across the entire softmax.
>
> : Dropping class logits (o_t = 0) does not raise the magnitude of logits of negative classes. Rather, it is equivalent to setting class probabilities to neutral (p_t = 1/T), which is “neither certainly positive(+) nor negative(-)” for a given instance. However, we corrected it by setting exp(o_t)=0 to completely exclude a class from classification boundary as suggested.

---

### Public Comment · (anonymous) · 2017-12-16
**Good Results, but needs more clarity.**

Rating: 7 It does produce a better result
Review: Our review is based on reproducibility

Overall the paper seemed to reasonably comply with the standards of reproducibility set out for this challenge. The data was very easily obtainable and its partitions were well deﬁned. The DropMax paper mentioned what frameworks were used but did not give any code or pseudo-code. The DropMax paper’s hyperparameter selection for the adaptive DropMax model was well stated but was non-existent for the other models. Despite this lack of documentation, we were able to show the distinct improvement DropMax had over the base networks. The language of the DropMax paper was clear but could have been made drastically more clear by including a diagram of the network the paper was proposing. The DropMax paper did not mention any of the computing hardware used. The runtime of the experiments was signiﬁcant but reasonable for an academic research setting. Based on the above compliance with the criteria of reproducibility set forth by Joelle Pineau we believe that DropMax is adequately reproducible. We give it a 7/10 overall on reproducibility.


As other commentators pointed out, DropMax, as proposed in the paper, drops class logits and not classes. This is the cause of z being identically 1 without regularization of rho; if o_t is negative for some non-target class, then ∂L/∂ρt
is positive since dropping the logit for that class actually increases the predicted
the probability for that class. We confirm that this is the cause of failure experimentally
by applying ReLU activation to o_t, and successfully avoid z = 1 without having to use
regularization on ρ. However, our validation experiments show that regularization of ρ
can still be helpful, even when dropping classes.

https://github.com/jamesal1/DropMax.git

Confidence: 4 We are basing our opinions on the code we used to copy the results.

---

> ### Author Response · Authors · 2018-01-05
> **Clarification**
>
> We really appreciate your effort on reproduction of the experimental results. Here we clarify what you have mentioned about the experimental setup.
>
> 1.     Experimental setup for Cifar10 and 100: The batch size is 128, and the number of epoch is 200. Weight decay is fixed at 1e-4. Learning rate starts from 0.1 and multiplied by 0.1 at 80, 120, and 160-th epoch. We used SGD optimizer with momentum of 0.9. The baseline model is resnet-34, which you can obtain from https://github.com/tensorflow/models/tree/master/official/resnet.
>
> 2.     Experimental setup for AwA: The batch size is 125 and the number of epoch is 100. Weight decay is fixed at 1e-4. Learning rate starts from 0.001 and is multiplied by 0.1 at 30 and 60 epochs. We used the SGD optimizer with the momentum of 0.9. You can obtain the pretrained model and code from https://github.com/kratzert/finetune_alexnet_with_tensorflow, with explanation from https://kratzert.github.io/2017/02/24/finetuning-alexnet-with-tensorflow.html.
>
> 3.     We used the same validation set for tuning of hyperparameters for all other models.
>
> 4.     S = 100 in MNIST, while S=30 for other dataset. However, we do not consider S as a significant factor.
>
> 5.     We updated the convergence plot in the revision.
>
> 6.     The variational term is essential for valid variational inference, and thus it should not be ignored. We checked it with our own experimental setting, that the variational term is also crucial for the performance.
>
> 7.     Instead of dropping out logits, in our revision, we drop out class exponentiation as you have suggested.

---

### Author Response · Authors · 2018-01-05
**Summary of updates in the revision**

We really appreciate the constructive comments from all reviewers and thank to the UC Irvine team for reproduction of the experimental results. Here we briefly mention what has been updated in the revision. For more detailed explanations, please refer to the response to each reviewer.

1.     Instead of dropping out class logits, exponentiations of logits are dropped in the revision, as suggested by AnonReviewer3.
2.     All the experimental results, corresponding figures and Dropmax contours are updated according to the change in 1 -- dropping out the exponentiations.
3.     We added Epsilon in Eq. (3) to prevent the denominator from becoming zero.
4.     We added Figure 1 to illustrate the concept of how Dropmax improves fine-grained recognition.
5.     We added in deterministic attention baseline, as suggested by AnonReviewer3.
6.     We updated the learning curve (Figure 3). Now it is more stable and easy to interpret.

---

### Decision · Program_Chairs · 2018-01-29
**ICLR 2018 Conference Acceptance Decision**

**Decision:**

Invite to Workshop Track

**Comment:**

This paper proposes a general regularization algorithm which builds on the dropout idea. This is a very significant topic. The overall motivation is good, but the specific design choices are less well motivated over, for example, ad-hoc choices. Some concerns remain after the post-rebuttal discussion with the reviewers: the improvement is incremental in terms of concepts and methodology, the clarity needs to be improved and the experiments are somehow weak.
In summary, the main idea and research direction is interesting, but the attempted generality of the algorithm and the significance of the area call for a more clear and convincing presentation.